# Probing ultrafast spin-relaxation and precession dynamics in a cuprate Mott insulator with seven-femtosecond optical pulses

T. Miyamoto[1], Y. Matsui[1], T. Terashige[2], T. Morimoto[1], N. Sono[1], H. Yada[1], S. Ishihara[3], Y. Watanabe[4], S. Adachi[4], T. Ito[5], K. Oka[5], A. Sawa[5] & H. Okamoto [1,2]

A charge excitation in a two-dimensional Mott insulator is strongly coupled with the surrounding spins, which is observed as magnetic-polaron formations of doped carriers and a magnon sideband in the Mott-gap transition spectrum. However, the dynamics related to the spin sector are difficult to measure. Here, we show that pump-probe reflection spectroscopy with seven-femtosecond laser pulses can detect the optically induced spin dynamics in $Nd_2CuO_4$, a typical cuprate Mott insulator. The bleaching signal at the Mott-gap transition is enhanced at ~18 fs. This time constant is attributable to the spin-relaxation time during magnetic-polaron formation, which is characterized by the exchange interaction. More importantly, ultrafast coherent oscillations appear in the time evolution of the reflectivity changes, and their frequencies (1400–2700 cm$^{-1}$) are equal to the probe energy measured from the Mott-gap transition peak. These oscillations can be interpreted as the interference between charge excitations with two magnons originating from charge–spin coupling.

[1] Department of Advanced Materials Science, University of Tokyo, Chiba 277-8561, Japan. [2] AIST-UTokyo Advanced Operando-Measurement Technology Open Innovation Laboratory (OPERANDO-OIL), National Institute of Advanced Industrial Science and Technology (AIST), Chiba 277-8568, Japan. [3] Department of Physics, Tohoku University, Sendai 980-8578, Japan. [4] Department of Chemistry, Kyoto University, Kitashirakawa Oiwake-cho, Sakyo-ku, Kyoto 606-8502, Japan. [5] National Institute of Advanced Industrial Science and Technology, Tsukuba 305-8565 Ibaraki, Japan. Correspondence and requests for materials should be addressed to H.O. (email: okamotoh@k.u-tokyo.ac.jp)

Carrier doping induces Mott-insulator to metal transitions in various transition-metal oxides[1]. Typical examples are layered cuprates such as $La_2CuO_4$ and $Nd_2CuO_4$[2–6]. In such cuprates, charge–spin coupling strongly affects the dynamics of charge carriers on the antiferromagnetic spin background[7–10]. Theoretical studies predict that an isolated carrier forms a kind of magnetic polaron via spin-orientation changes[11–14] and that the Mott-gap transition peak in the optical conductivity ($\sigma$) spectrum is accompanied by a two-magnon sideband via charge–spin coupling[15]. However, information about spin dynamics associated with the magnetic-polaron formation and the two-magnon excitation has yet to be observed. Femtosecond pump–probe (PP) spectroscopy may address these unsolved issues since it can detect the time characteristics of photoexcited states[16–21], helping to derive spin-dynamics information.

The studied material is $Nd_2CuO_4$. The Cu–O plane of $Nd_2CuO_4$ is schematically shown in Fig. 1. The copper $3d_{x^2-y^2}$ and oxygen $2p_{x,y}$ orbitals form the two-dimensional electronic state. In the copper $3d$ band, the Mott–Hubbard gap is opened due to the large on-site Coulomb repulsion. The occupied oxygen $2p$ band is located between the copper $3d$ upper- and lower-Hubbard bands. The lowest electronic excitation is the charge-transfer (CT) transition from the oxygen $2p$ band to the copper $3d$ upper-Hubbard band. Hereafter, the CT transition is simply called the Mott-gap transition. The strong $pd$ hybridization leads to a large antiferromagnetic exchange interaction $J \sim 0.155$ eV between two neighboring copper spins[22].

When electron carriers are introduced into $CuO_2$ planes by substituting Nd with Ce in $Nd_{2-x}Ce_xCuO_4$, the spectral weight of the Mott-gap transition is transferred to the mid-gap absorption[2–4] originating from magnetic polarons, and the Drude component is enhanced for $x > 0.1$. However, the Drude weight is small compared to the mid-gap absorption even in optimally doped samples exhibiting superconductivity[3,4]. In $Nd_2CuO_4$, the photocarrier dynamics have been investigated by PP absorption spectroscopy with time resolutions of 40–200 fs[16,17]; a photo-induced Mott-insulator to metal transition was demonstrated, but the spin-relaxation processes of photocarriers were not detected due to the insufficient time resolution. Note that the time scale of the spin dynamics evaluated from $J \sim 0.155$ eV is ~30 fs.

In this study, we perform PP reflection spectroscopy on $Nd_2CuO_4$ single crystals with ultrashort pulses with the temporal width of 7 fs to detect spin dynamics after photo-excitation. In the weak excitation case, photocarriers form magnetic polarons in ~18 fs. With the increase of excitation photon density, Drude components are enhanced and Mott insulator to metal transition occurs. Remarkably, measured reflectivity changes include high-frequency oscillations whose frequencies depend on probe energy. The theoretical calculation reveals that those oscillations originate from the quantum interference between the optical transitions related to the two-magnon sideband.

## Results

**Experimental setup.** Figure 2a shows the reflectivity ($R$) and $\sigma$ spectra of $Nd_2CuO_4$ in which the Mott-gap transition is observed as the peak structure. A schematic of the experimental setup is illustrated in Fig. 2b. Pump and probe pulses are generated from a handmade non-collinear optical parametric amplifier. The cross-correlation profile of the pump and probe pulses is shown in Fig. 2c. The temporal width of this profile, which is about 10 fs, corresponds to the time resolution in our system. This pulse is divided into two pulses, which are used as pump and probe pulses. The spectrum of the pulse is shown by the colored area in Fig. 2a, which is located within the Mott-gap-transition band. By

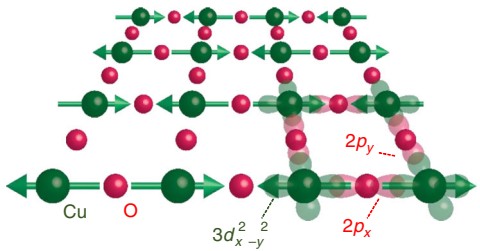

**Fig. 1** Schematic of the $CuO_2$ $ab$ plane. The electronic spin arrangement is illustrated with green arrows. The translucent lobes represent the electronic orbitals that form the two-dimensional electronic state

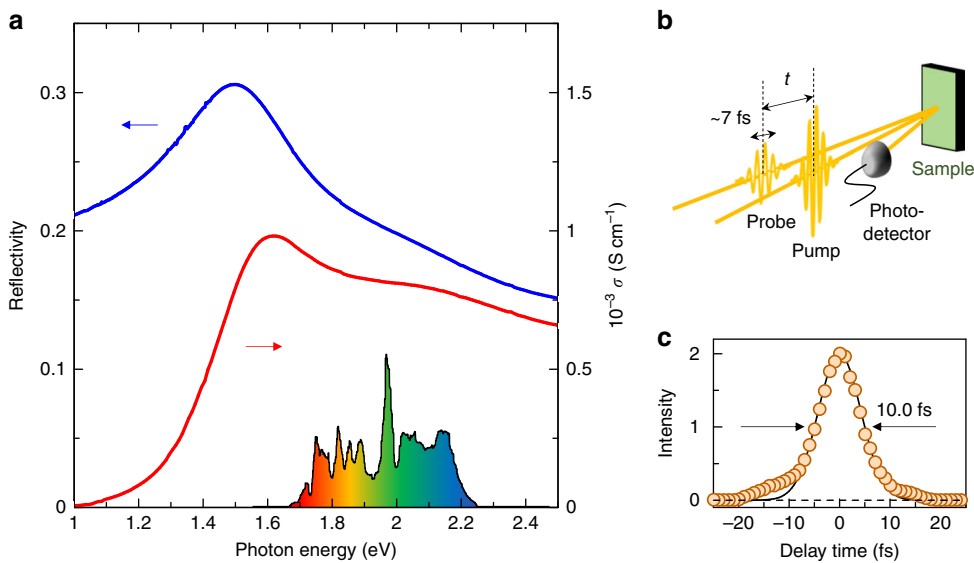

**Fig. 2** Optical spectra and pump-probe reflection setup on $CuO_2$ planes in $Nd_2CuO_4$. **a** The reflectivity ($R$, blue) and optical conductivity ($\sigma$, red) spectra measured on the $ab$ plane. Electric fields of lights are polarized parallel to $a$ or $b$ axis. The black line in the lower part shows a spectrum of a 7-fs pulse used in the pump-probe (PP) experiments. **b** A schematic of PP experiments. **c** A cross-correlation profile of pump and probe pulses corresponding to the time resolution

detecting the transient decrease in the reflectivity of the probe pulse, that is, the bleaching signal, we can investigate the photocarrier dynamics.

**Excitation-photon-density dependence of the reflectivity changes**. Figure 3a shows the time evolutions of the reflectivity changes $\Delta R(t)/R$ of the probe pulse for the excitation photon density $x_{ph} = 0.0016$–$0.079$ ph/Cu. The cross-correlation profile corresponding to the time resolution is also shown by the yellow line in Fig. 3a. Photoirradiation decreases the reflectivity due to photocarrier generation. This bleaching signal reflects the dynamics of both the mid-gap-absorption and Drude components, since the spectral weight of the Mott-gap transition is transferred to the latter two components[16,17]. Figure 3b–d shows typical time characteristics of $\Delta R(t)/R$ for weak ($x_{ph} = 0.0049$ ph/ Cu), medium ($x_{ph} = 0.024$ ph/Cu), and strong ($x_{ph} = 0.079$ ph/ Cu) excitations, respectively. In the weak-excitation case, $\Delta R(t)/R$ initially decreases within the time resolution (step A), further decreases with a rise time of ~20 fs (step B), and is followed by a slow decay of ~350 fs (step C). In $Nd_{2-x}Ce_xCuO_4$ with $x \leq 0.025$, a metallic behavior is not observed and the system is semiconducting[3,4]. Therefore, it is natural to consider that step B is related not to the Drude component but to the magnetic-polaron formations.

In the strong-excitation case (Fig. 3d), $\Delta R(t)/R$ reaches a minimum just after photoirradiation (step A). Subsequently, part of $\Delta R(t)/R$ decays within ~30 fs (step B') and the residual part

decays within ~350 fs similar to the weak excitation case (step C). The former ultrafast-decay component gradually increases with increasing $x_{ph}$ (Fig. 3a) and is attributed to the Drude component, which is generated from the overlapping wavefunctions of electron (hole) carriers. In the negative time region ($t < 0$), the $\Delta R(t)/R$ signal appears at $t = -100$ fs and grows as $t$ approaches 0. This can be ascribed to the free induction decay (FID) or the reduction of probe-pulse-induced polarization by the pump pulse[23] (Supplementary Note 1). This component is not related to the photocarrier dynamics discussed here.

**Analyses of the time evolutions of the reflectivity changes**. To analyze the time evolution of $\Delta R(t)/R$, the following formula is adopted in the weak excitation case.

$$\frac{\Delta R(t)}{R} = -\left\{ A_1 + A_2 \left[ 1 - \exp\left(-\frac{t}{\tau_r}\right) \right] \right\} \exp\left(-\frac{t}{\tau_1}\right) \quad (1)$$

The first and second terms show bleaching due to the initial photocarrier generation (step A) and the formation of the mid-gap states with time constant $\tau_r$ (step B), respectively. $\tau_1$ is the decay time of the mid-gap states (step C). $\Delta R(t)/R$ for $x_{ph} = 0.0049$ ph/Cu can be well reproduced by Eq. (1) with $\tau_r = 18$ fs as shown by the solid line in Fig. 3b, except for the FID components.

First, we discuss the origin of the reflectivity change in step B, since it is an unusual dynamics. As mentioned above, it is natural to relate this step B to the magnetic-polaron formation, while

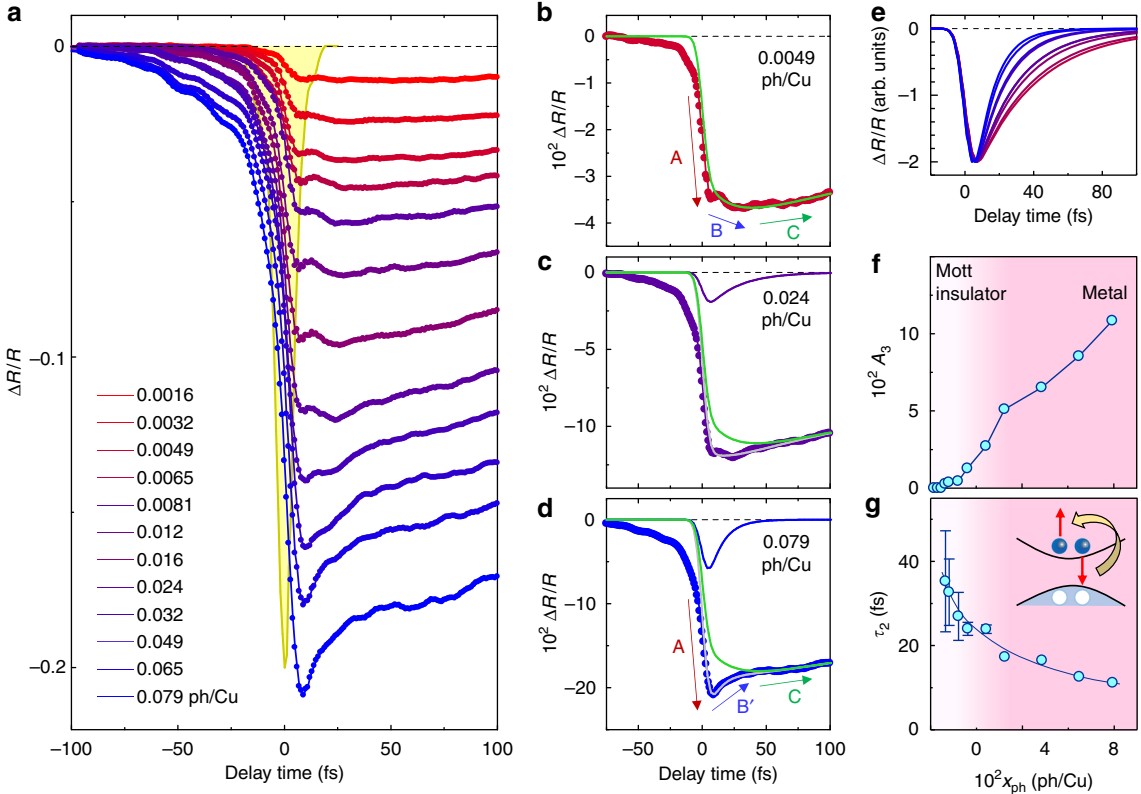

**Fig. 3** Results of the pump-probe reflection measurements in $Nd_2CuO_4$. **a** Time evolutions of reflectivity changes $\Delta R/R$ for $x_{ph} = 0.0016$–$0.079$ ph/Cu. A yellow line reshows the cross-correlation profile between pump and probe pulses. **b–d** Time evolutions of $\Delta R/R$ for **b** $x_{ph} = 0.0049$ ph/Cu, **c** $x_{ph} = 0.024$ ph/Cu, and **d** $x_{ph} = 0.079$ ph/Cu with fitting curves. Green and purple lines in (**c**) (green and blue ones in (**d**)) reflect the mid-gap-absorption components and the Drude components, respectively. Thin gray lines show the sum of two components. **e** Time evolutions of reflectivity changes $\Delta R/R$ reflecting the Drude components for $x_{ph} = 0.0065$–$0.079$ ph/Cu derived from the fitting analyses. The colors are the same as in (**a**). **f**, **g** $x_{ph}$-dependences of **f** amplitudes and **g** decay time of the reflectivity changes $\Delta R/R$ reflecting the Drude components. Error bars in (**g**) show standard deviations of the least-squares fit. The inset shows a schematic illustration of the Auger recombination

charge–phonon coupling should also be considered as well as charge–spin coupling in the formation of magnetic polaron. In fact, in the previous study of the photoemission spectroscopy on another typical undoped cuprate $La_2CuO_4$, a polaronic behavior of a hole was identified[24]. However, we can consider that the charge–phonon coupling does not play important roles on step B observed in $Nd_2CuO_4$ from the following reasons. The sign of the additional reflectivity change in step B is negative and the same as that of the initial decrease of the reflectivity (step A), that is, the main bleaching signal due to the photocarrier generation. Namely, in step B, the bleaching signal or the absolute value of the negative reflectivity change $|\Delta R/R|$ is enhanced up to $t \sim 20$ fs. This means that photocarriers become more itinerant with the rise time of $\tau_r = 18$ fs. If photocarriers are relaxed to small polarons due to the charge–phonon coupling (lattice polarons) in the same time domain, the itinerancy of carriers is suppressed by the surrounding displacements of ions. This should cause the reduction of the bleaching signal, resulting in the decrease of the absolute value of the negative reflectivity change $|\Delta R/R|$. The experimental result (step B) shows the opposite behavior, that is, the further decrease of the reflectivity. Therefore, the lattice-polaron formation cannot explain the experimental results. This interpretation is also supported by the timescale of optical phonons. The highest frequency (energy) of the phonon dispersion curves in $Nd_2CuO_4$ is ~18 THz (74 meV)[25]. The period of that mode is ~56 fs, which is much longer than the time constant $\tau_r = 18$ fs in the response of step B. This fact also excludes the contribution of the charge–phonon coupling to step B. An introduction of a photocarrier itself necessarily relaxes the surrounding antiferromagnetic spin arrangement. That makes the carriers a bit more itinerant and therefore leads to the increase of the bleaching signal as observed in the experiments. Thus, we can conclude that the magnetic-polaron formation is responsible for step B, and $\tau_r = 18$ fs is the spin-relaxation time.

In our results, no dynamical behavior originating from the charge–phonon coupling is observed. This is attributable to the fact that the charge–phonon coupling in $Nd_2CuO_4$ is relatively smaller than in $La_2CuO_4$, in which the effect of the charge–phonon coupling on doped carriers was demonstrated. We consider that in $Nd_2CuO_4$, the charge–phonon coupling is relatively smaller than in $La_2CuO_4$ mainly because apical oxygen atoms are absent in $Nd_2CuO_4$. The role of apical oxygen atoms on the charge–phonon coupling was detailed in the previous studies[16,17].

To analyze $\Delta R(t)/R$ in the medium and strong excitation case (Fig. 3c, d, respectively), we add the term $-A_3 \exp(-t/\tau_2)$ to Eq. (1) to express the Drude component with decay time $\tau_2$ (step B'). In the analyses, we use the same values of $\tau_r$ and $A_2/A_1$ as those obtained for the weak excitation case. The time characteristic of $|\Delta R(t)/R|$ for $x_{ph} = 0.024$ and 0.079 ph/Cu are well reproduced as shown by gray lines in Fig. 3c, d. The Drude and mid-gap-absorption components are shown by purple and green lines in Fig. 3c (blue and green lines in Fig. 3d), respectively.

$|\Delta R(t)/R|$ for various $x_{ph}$ values can also be reproduced by the same formula (Supplementary Note 2). The $|\Delta R(t)/R|$ signals reflecting the Drude components thus derived are shown in Fig. 3e. The values of the parameters $A_3$ and $\tau_2$, which reflect the effective number of free carriers and their decay times, are plotted as a function of $x_{ph}$ in Fig. 3f, g, respectively. The effective number of free carriers or the spectral weight of the Drude component ($A_3$) shows a clear threshold behavior, which is characteristic of two-dimensional Mott insulators[17]. As the carriers contributing to the Drude component increase, their decay time $\tau_2$ decreases. These behaviors suggest that an Auger recombination[26] illustrated in the inset of Fig. 3g plays a dominant role on the ultrafast decay of the metallic state. Detailed analyses of the long-time data ($t > 10$ fs) reveal that when

the carrier number becomes small, the residual electron and hole carriers recombine with $\tau_1 \sim 350$ fs (Supplementary Note 3), which is thought to occur via the theoretically predicted multi-magnon emission[27,28].

In the analyses for the medium- and strong-excitation cases, we used the same parameter values of $\tau_r$ and $A_2/A_1$ showing the time and the magnitude of the spin relaxation as mentioned above. This is a crude assumption, because with increase of $x_{ph}$ the antiferromagnetic spin arrangement is further disturbed and $\tau_r$ might become long. In the analyses of the medium- and strong-excitation cases, however, it is important to evaluate the excitation photon density dependence of the magnitude $A_3$ and the decay time $\tau_2$ of the Drude component. Even if the spin relaxation time $\tau_r$ becomes longer with increase of $x_{ph}$, the characteristic features of the $x_{ph}$ dependence of $A_3$ and $\tau_2$ shown in Fig. 3f, g, respectively, would not be affected so much, since the Drude component shows apparently different behaviors from the spin relaxation processes of the magnetic polarons as shown in Fig. 3b–d.

**Probe-energy dependence of the reflectivity changes**. To derive the detailed information about the magnon sideband originating from the charge–spin coupling, we next measured the probe-energy dependence on $\Delta R(t)/R$ (see Methods). Figure 4a shows the time evolution of $\Delta R(t)/R$ for the four probe energies (1.88, 1.94, 2.03, and 2.10). The excitation photon density is $x_{ph} = 0.008$ ph/Cu. This corresponds to the weak excitation case in which the Drude component can be neglected. All the time profiles for the four probe energies contain the high-frequency oscillatory structures. We extracted those oscillatory components $\Delta R_{OSC}(t)/R$ from $\Delta R(t)/R$ using a Fourier filter, which are plotted in Fig. 4b. The Fourier power spectra of $\Delta R_{OSC}(t)/R$ are also shown in Fig. 4c. The oscillation frequencies ($\hbar\Omega_n$) depend on the probe energy ($\hbar\omega_r$), which is an unusual behavior. Figure 4d plots $\hbar\Omega_n$ as a function of $\hbar\omega_r$ together with the $\sigma$ spectrum. $\hbar\Omega_n$ increases from the Mott-gap transition peak up to 2700 cm$^{-1}$, following the relation $\hbar\Omega_n = \hbar\omega_r - 1.74$ eV, which is shown by the broad gray line in Fig. 4d. 1.74 eV is close to the Mott-gap transition energy $\hbar\omega_{CT}$. This suggests that the oscillations may be related to the emission of two magnons.

The amplitudes of the oscillations are also expected to include important information about the charge–spin coupling. However, in our experimental condition, it is difficult to compare precisely the relative oscillation amplitudes detected at different probe energies, since the oscillation frequencies are very high. The period of the oscillation with the frequency of 2700 cm$^{-1}$ observed at the probe energy of 2.10 eV is 12.4 fs, which is comparable to the time resolution of 10 fs in our PP system. In such a situation, the amplitudes of the oscillation experimentally measured are suppressed with an increase of the oscillation frequency. On the other hand, the central frequency of each oscillatory component can be accurately determined directly from the experimental data. So, we focused on it in the present study. The origin of the observed oscillations is discussed in detail in the next section.

## Discussion

Here, we discuss the origin of the high-frequency coherent oscillations observed on the reflectivity changes after the photo-irradiation. In cuprate Mott insulators, two-magnon emission signatures are observed in the Raman scatterings[29]. It is therefore natural to consider that the two-magnon sideband is included on the higher-energy side of the Mott-gap transition peak. Figure 4e, f illustrates this sideband and the magnon dispersion[30], respectively. In the emission of two magnons with positive and negative momentums, the total spin is conserved. Thus, the two-magnon frequency $\hbar\Omega_n$ is connected to the probe energy $\hbar\omega_r$ via $\hbar\omega_r = \hbar\omega_{CT} + \hbar\Omega_n$ as observed in the experiments.

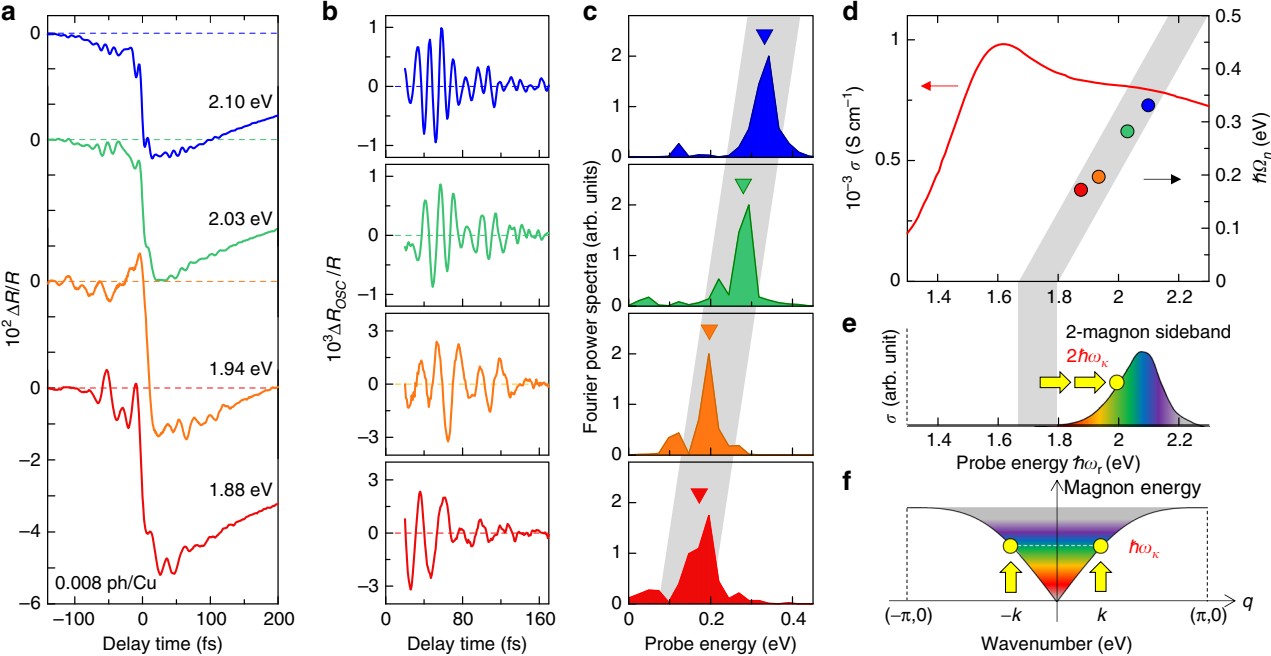

**Fig. 4** Probe energy dependence of reflectivity changes in $Nd_2CuO_4$. **a** Time evolutions of $\Delta R/R$ for $x_{ph} = 0.008$ ph/Cu. The red, orange, green, and blue colors show the data at the probe energies ($\hbar\omega_r$) of 1.88, 1.94, 2.03, and 2.10 eV, respectively, in (**a–d**) in common. **b** Oscillatory components $\Delta R_{OSC}/R$ extracted from $\Delta R/R$. **c** Fourier power spectra of $\Delta R_{OSC}/R$. Triangles indicate the oscillation frequencies $\hbar\Omega_n$. **d** Probe energy dependence of $\hbar\Omega_n$. The solid line shows the $\sigma$ spectrum. The broad gray line is drawn from 1.73 eV ($\hbar\omega_{CT}$) with the slope of 1 (see the text). **e** A two-magnon sideband is schematically shown as the shaded area. **f** Schematics of the magnon dispersion

To explain the generation of the coherent oscillations, we consider the photoexcited states $|N,n\rangle$, where $N$ ($=1$ or 2) denotes the number of electron (doublon)–hole (holon) pairs and $n$ denotes the two-magnon excited state with the frequency $\hbar\Omega_n$. A broadband pump pulse generates $|1,n\rangle$ with various frequencies $\hbar\Omega_n$ and a probe pulse produces additional transitions (e.g., $|1,0\rangle \rightarrow |2,n\rangle$ and $|1,n\rangle \rightarrow |2,n\rangle$), which interfere with each other. This results in a coherent oscillation with the frequency $\hbar\Omega_n$, which corresponds to the energy difference between the initial photoexcited states $|1,0\rangle$ and $|1,n\rangle$. The coherent oscillation should be detected at the probe energy of $\hbar\omega_r = \hbar\omega_{CT} + \hbar\Omega_n$, which is the energy for $|1,n\rangle$. The results and the theoretical simulation of the oscillations are reported in Supplementary Note 4.

Finally, we summarize the photocarrier relaxation processes in $Nd_2CuO_4$, which is schematically shown in Fig. 5. In the weak excitation case, photocarriers form magnetic polarons in ~18 fs ($a \rightarrow b \rightarrow d$ in Fig. 5). In the strong excitation case, a number of carriers are generated just after photoirradiation ($a \rightarrow c$ in Fig. 5). The system becomes metallic because the wavefunctions of the photocarriers overlap and the metallic state decays in ~10–40 fs ($c \rightarrow d$ in Fig. 5), depending on the initial photocarrier densities. An Auger recombination is a plausible decay process. The residual small numbers of photocarriers form magnetic polarons. The coherent oscillations observed in the pump–probe responses are attributed to quantum interferences between charge–spin coupled excited states.

In summary, we have applied pump–probe reflection spectroscopy with a time resolution of 10 fs to a typical Mott insulating undoped cuprate, $Nd_2CuO_4$. We observed ultrafast responses originating from the charge–spin coupling in the photoexcited state. In the weak excitation case, we clearly identified the enhancement of the bleaching signal with a rise time of 18 fs just after the photocarrier generation, which was attributed

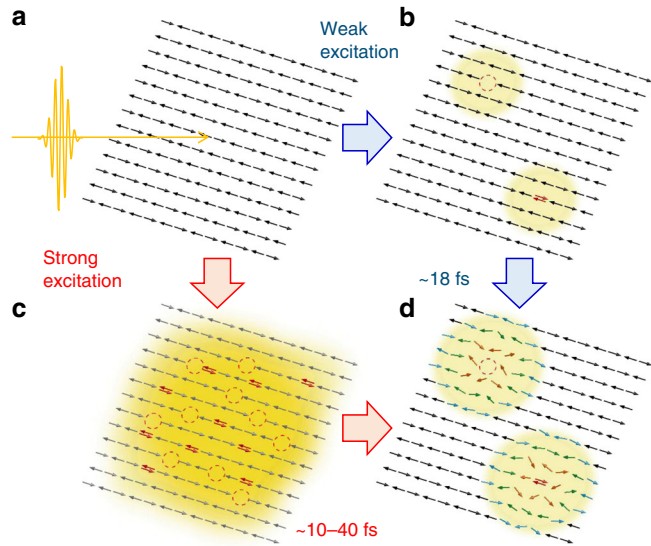

**Fig. 5** A schematic of the ultrafast photoresponses in $Nd_2CuO_4$. **a** The Mott insulator state. **b, c** Generation of **b** low density and **c** high density of doublons and holons just after the photoirradiation. **d** Formations of magnetic polarons in the relaxation process

to the time constant of the magnetic-polaron formation. In the spectrally resolved pump-probe experiments, we detected a high-frequency (1400–2700 cm⁻¹) coherent oscillation in the reflectivity changes. The oscillation frequency depends on the probe energy, which was explained as the interference between charge excitations with two magnons. This is not only evidence that the charge–spin coupling generates a two-magnon sideband just

above the Mott-gap transition peak but also the first real-time detection of charge–spin coupled dynamics in a Mott insulator of undoped cuprates. The results presented here are a benchmark that can be compared to modern theoretical calculations of non-equilibrium charge and spin dynamics in photoexcited Mott insulators.

## Methods

**Sample preparations.** $Nd_2CuO_4$ single crystals were grown in air by a modified self-flux method[31]. A mixture of 200 g of $Nd_2O_3$ and CuO at a molar ratio of Nd: Cu = 28:72 was filled in a Pt crucible. The crucible was heated at 1300 °C for 2 h in a furnace to melt the mixture completely. Then the tip of a Pt wire was dipped in the melt, and the temperature was lowered at a speed of 3 °C/h to 1220 °C. The crystals grown around the tip were raised by pulling up the wire. The melt around the crystals was dripped, forming plate-like crystals with shiny flat surfaces.

**Steady-state optical spectroscopy measurements.** The optical reflectivity (R) spectrum in a single crystal of $Nd_2CuO_4$ was measured using a specially designed spectrometer with a 25-cm-grating monochromator for 0.5–5.0 eV and a Fourier-transform infrared spectrometer for 0.15–1.2 eV. Both were equipped with an optical microscope. The optical conductivity ($\sigma$) spectrum was calculated from the R spectrum by the Kramers–Kronig transformation.

**Femtosecond pump–probe reflection measurements.** The light source for the PP measurements was a Ti:sapphire regenerative amplifier with a central wavelength of 800 nm, pulse width of 130 fs, pulse fluence of 0.8 mJ, and repetition rate of 1 kHz. To generate an ultrashort pulse, we self-produced a non-collinear optical parametric amplifier[32,33]. We divided the output of the regenerative amplifier into two beams. One was converted to second harmonic (SH) light using a $LiB_3O_5$ crystal. The other was focused on a sapphire crystal, from which a broadband white-light pulse was generated. This white-light pulse was used as a seed and amplified by the SH light in a $\beta$-$BaB_2O_4$ crystal. The pulse width at the sample position was compressed by a pair of chirp mirrors and quartz plates inserted between the sapphire crystal and the sample. Finally, an ultrashort pulse with a temporal width of 7 fs and a spectrum width of ~0.5 eV (Fig. 2a) was obtained.

The setup of the PP reflectivity measurements is illustrated in Fig. 2b. The 7-fs pulse was divided into two pulses, which were used as pump and probe pulses. The polarization of the pump and probe pulses was the same and parallel to either the a or b axis. A variable delay stage controlled the delay time of the probe pulse relative to the pump pulse. To suppress the effects of the intensity fluctuations of the probe pulse on the signals, we normalized the reflection intensity of each probe pulse by the intensity of the pulse before the incidence, which was measured by extracting part of the probe pulse using a semitransparent mirror. The cross-correlation profile of the pump and probe pulses, which is shown by the yellow line with the yellow shading in Fig. 3a. The full width at the half maximum was 10.0 fs, corresponding to the time resolution. In the measurements of the probe-energy dependence of the reflectivity changes, we selected part of the reflection light by inserting a band-pass filter (bandwidth of 10 nm and central wavelengths of 588, 610, 636, or 656 nm) in front of the detector (a photodiode).

The excitation photon density per Cu site, $x_{ph}$, was evaluated from the formula $x_{ph} = I_p(1 - R_p)(1 - 1/e)/l_p$, in which $I_p$, $R_p$, and $l_p$ are the photon density per unit area, reflectivity, and penetration depth of the pump pulse, respectively. $l_p$ was evaluated from the absorption-coefficient spectrum, which was obtained by the Kramers–Kronig analysis of the R spectrum.

## Data availability

The data that support the plots within this paper and other findings of this study are available from the corresponding author upon reasonable request.

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

## Acknowledgements

We thank Mr. Y. Miyata, Mr. N. Osawa, and Mr. M. Inoue for their collaborations in the early stage of this study. This work was partly supported by Grants-in-Aid for Scientific Research from the Japan Society for the Promotion of Science (JSPS) (Project Numbers:

JP15H02100, JP15H06130, JP16K17721, and JP17H02916) and by CREST (Grant Number: JPMJCR1661), Japan Science and Technology Agency. T.T. and T. Morimoto were supported by the JSPS through the Program for Leading Graduate Schools (MERIT).

## Author contributions

T. Miyamoto, Y.M., T. Morimoto, N.S., Y.W., and S.A. built the experimental apparatus. T. Miyamoto., Y.M., T.T., N.S., and H.Y. carried out the transient optical measurements. T.I., K.O., and A.S. prepared the samples. S.I. performed theoretical calculations. H.O. coordinated the study. All of the authors discussed the results and contributed to writing the paper.

## Additional information

**Competing interests:** The authors declare no competing interests.

