## [Peer Review File · Nature Communications]

Reviewers' comments:

Reviewer #1 (Remarks to the Author):

Miyamoto et al. report ultra-fast optical reflectivity studies of cuprate Mott insulator Nd₂CuO₄. The main conclusion (which I will call conclusion A) of the manuscript is that there is a spin relaxation time of 18 fs associated with spin-polaron formation. It also reports (Conclusion B) oscillations of the reflectivity above the gap, which are assigned to two-magnon creation.

The research team achieved an outstanding time resolution of 10 fs, which is a major strength of the study. It is also reasonable to say that the behavior of Mott insulating cuprates is very important because they are often thought of as the canonical strongly correlated problem. The combination of these points put this study in good contention for Nat. Comms. Having said this, I think that Conclusion A is not sufficiently strongly established to qualify for Nat. Comms. Conclusion B is well-established, but I do not think there is a sufficiently strong case that something important about cuprates or ultra-fast optics has been learned here. I would be willing to re-assess this if the authors can expand on a stronger argument for conclusion A or a more compelling discussion of the importance of conclusion B.

In order to start this conversation, I would emphasize that cuprates are well known to have strong charge-spin coupling and are very often discussed as having appreciable electron-phonon coupling. So one really needs to add detailed information about these processes rather than just show this as a general truth.

Conclusion A:

The authors adequately demonstrate that the optical reflectivity of Nd₂CuO₄ contains a dynamical component with a timescale of 18 fs. I am struggling to establish the exact logic connecting this to "spin relaxation". My understanding is that the assertions are:

- Lightly doped cuprates are insulating due to "spin-polaron" effects up to a static carrier doping of ~2%
- The reflectivity effect seems to change upon an effective transient doublon concentration of about 2% associating the 18 fs timescale with the insulating transition and the "spin-polarons"

My main problem with this line of reasoning is that I think it's incorrect to present the transition as purely "spin-polaron". There can (and seemingly must) be a polaron associated with charge effects too. This is discussed a lot in the literature [e.g. PRL 95, 227002 (2005)]. While I agree that coherent long wavelength phonon effects might be slower than 18 fs, I don't see why local polaronic charge effects can't be very fast.

I would also note that reflectivity is fundamentally a probe of charge, so in the case of mixed character polarons, charge effects might be more likely to be seen.

Conclusion B:

I think this effect is nicely demonstrated. This not immediately obvious to me whether anything new has been learned regarding the cuprates. I would be interested in any further arguments or developments of the narrative in manuscript.

Small comments:

1. Page 3: this sentence implies the charge carriers are moving, "In such cuprates, charge-spin coupling affects a charge carrier strongly, which moves on the antiferromagnetic spin background⁷⁻¹⁰."

A major point of the article is the spin-polaron effect, in which the carriers do not move.

2. Figure 1 is titled: "pump-probe reflection measurements", so I spent a few seconds looking for a pump-probe measurement, but what is meant here is pump-probe reflection setup
3. Figure 1 a) the spins are in the wrong direction.
4. In Fig 3f, maybe there is an unusual unit choice, but I usually expect to see the maximum of the dispersion at $(\pi, 0)$.

Reviewer #2 (Remarks to the Author):

The manuscript by Miyamoto et al. presents an experimental investigation of the relaxation dynamics of photo-induced carriers in an undoped cuprate Mott insulator. A 7fs/2eV laser pulse is used to excite carriers across the gap and probe the subsequent bleaching of the charge-transfer (CT) resonance. The dynamics is analyzed in detail as function of the excitation density: For low photon number, an increase of the bleaching signal within 18fs after the excitation is observed, which is interpreted as a signature of the spin-polaron formation. When the fluence is increased, an additional fluence-dependent response is observed which reduces the bleaching; due to the fluence dependence, this can be interpreted as consequence of Auger recombination processes. Finally, an oscillatory component in the response at weak excitation density is interpreted as beating between CT-excitations with different magnon numbers.

A precise understanding of the carrier dynamics in Mott insulators is of great importance for the active research on materials out of equilibrium. The dynamics reflects the interaction of carriers and spins, which in turn is important for the superconducting pairing, so that a detailed knowledge of the ultra-fast carrier dynamics may help understanding light-induced superconductivity or superconductivity in the cuprates in general. In contrast to many other studies in the field, which focus on the doped material, the present work investigates the *undoped* cuprates. Only if the un-doped material is understood, there is a chance to shed light on the doped system. While the undoped material might not be as interesting regarding its material properties, I believe that the precise analysis as presented in the manuscript is a very important benchmark for the field: Because the properties of the undoped material are easier (but not easy!) to address theoretically, the study provides the basis for a detailed comparison to theory, which is crucially needed for further progress, but has rarely been attempted regarding the dynamics of strongly correlated systems in general. The observed timescale for spin-relaxation is in fact consistent with theoretical predictions (Ref. 7-10). Beyond that, the results of the manuscript can stimulate further studies: At present there is no systematic theoretical understanding including the excitation density dependence and hence the interaction between photo-doped carriers. Related open questions include the differentiation between p-holes and d-electrons and the p-d interaction. Furthermore, antiferromagnetic correlations in the 2d Hubbard model have recently been also studied in quantum simulation experiments with cold-atoms. The investigation of the short-time dynamics in the undoped cuprates seems to be a good candidate for a relatively clean setting in which a comparative study of experiments in these two fields may be envisaged. I therefore believe that the topic is timely and of broad interest and thus suitable for publication in Nature Communications.

Besides this I have a number of questions/comments regarding the discussion of the results:

- (1) It is possible to discuss the temperature-dependence of the results? Increasing temperature weakens the spin correlations and thus might have an effect on the relaxation process, which could give further support to the interpretation.
- (2) Why can it be assumed in the analysis of the strong-fluence data that the signal from the spin-polaron formation is not fluence dependent? One could expect a saturation effect, i.e., for strong fluence the spin order is partially melted, so that the relaxation should slow down. Is it possible to give an estimate at which excitation density such a saturation effect might be expected?

(3) The beating between the zero and two-magnon-dressed states (Fig.3) is a nice interpretation of the data. How much does it rely on the sharpness of the CT excitation? The CT resonance seems to be broader than the beating frequency. On the other hand, as the beating occurs at the difference frequency between probe frequency and the CT resonance, should it not de-phase on the timescale set by the inverse width of the CT resonance?

(4) Assuming that the present interpretation of the beating signal is correct, can one discuss what could be learned from the measured signal? First, its amplitude could give information on the density of states of the modes which are involved. For the two-magnon states, there might be both a maximal energy (set by the maximum of the magnon dispersion), but possibly also a minimal energy (if the material has no long-range antiferromagnetic order, there are just paramagnons which do not exist at arbitrary low energy). On the other hand, the amplitude of the signal does not seem to depend on the frequency (Fig 3c)? Second, the de-phasing of the beating may give information on the lifetime of the coherence between the CT and the CT+2magnon state. In this case, it may be interesting to discuss its dependence on the excitation density.

(5) For a further characterization of the beating signal it would be interesting to know how it scales with the amplitude of pump and probe. If it scales both linear in pump and probe in the weak excitation regime, it could be interpreted in terms of non-linear optics.

(6) This is a minor comment on the interpretation of process "B" in terms of spin-polaron formation: I agree that the interaction with lattice vibrations seems to be unlikely to be the origin of the process if the energy scale of optical phonons is smaller by a factor of three than the 220meV which correspond to 18fs. However, I think that in general such an argument should be used with care. The timescale on which energy is passed to some bosonic excitation is not simply related to the inverse of the energy of that bosonic excitation, but rather to the strength of its coupling to the electrons. The prime example is spin itself - the rate at which spin excitations are created in an antiferromagnet is set by the electron hopping, and the magnon energy only determines how much energy is transferred per spin excitation.

In summary, I find this is a nicely presented high-quality set of data which contribute to a very timely and important question in the field of ultra-fast dynamics in correlated systems. With a discussion that addresses the above points (as far as possible), I think the manuscript can be published in Nature Communications.

Response to Reviewers

Summary of changes

(1) Taking comment 2 of the reviewer #1 into account, we added the detailed description about the ultrafast dynamics in step B including the lattice polaron effect due to the charge-phonon coupling at line 9 in page 6 – line 19 of page 7. To make clear the discussion, we also slightly modified the related descriptions before the above discussions at line 1 – 4 in page 5 and at line 9 – 11 in page 5.

We also added the related reference in the revised manuscript.

24. Rösch, O. *et al.* Polaronic Behavior of Undoped High- T_c Cuprate Superconductors from Angle-Resolved Photoemission Spectra. *Phys. Rev. Lett.* **95**, 227002 (2005).

We renumbered the references in the revised manuscript.

(2) Taking comment 3 of the reviewer #1 into account, we isolated a summary paragraph and revised it at line 16 in page 11 – line 7 in page 12.

(3) Taking comment 4 of the reviewer #1 into account, we modified the sentence in the introduction at line 2 – 4 in page 3.

(4) Taking comment 5 of the reviewer #1 into account, we modified the title of Fig. 1.

(5) Taking comments 6 and 7 of the reviewer #1 into account, we corrected Figs. 1a, 3f and 4.

(6) Taking comment 2 of the reviewer #2 into account, we added the discussion about the analyses of the time evolutions of the reflectivity changes associated with the magnetic polaron and the Drude component at line 17 in page 8 – line 4 in page 9. In addition, we modified the related sentences at line 6 – 10 in page 8 and line 22 – 23 in page 7.

(7) Taking comment 4 of the reviewer #2 into account, we added a description about the analyses of the probe energy dependence of the oscillation amplitudes at line 21 in page 9 – line 9 in page 10.

These changes in the revised manuscript are marked in yellow in Marked MS.pdf.

Comments of the reviewer #1

Miyamoto et al. report ultra-fast optical reflectivity studies of cuprate Mott insulator Nd_2CuO_4 . The main conclusion (which I will call conclusion A) of the manuscript is that there is a spin relaxation time of 18 fs associated with spin-polaron formation. It also reports (Conclusion B) oscillations of the reflectivity above the gap, which are assigned to two-magnon creation.

The research team achieved an outstanding time resolution of 10 fs, which is a major strength of the study. It is also reasonable to say that the behavior of Mott insulating cuprates is very important because they are often thought of as the canonical strongly correlated problem. The combination of these points put this study in good contention for Nat. Comms.

Comment 1

Having said this, I think that Conclusion A is not sufficiently strongly established to qualify for Nat. Comms. Conclusion B is well-established, but I do not think there is a sufficiently strong case that something important about cuprates or ultra-fast optics has been learned here. I would be willing to re-assess this if the authors can expand on a stronger argument for conclusion A or a more compelling discussion of the importance of conclusion B.

Comment 2

In order to start this conversation, I would emphasize that cuprates are well known to have strong charge-spin coupling and are very often discussed as having appreciable electron-phonon coupling. So one really needs to add detailed information about these processes rather than just show this as a general truth.

Conclusion A:

The authors adequately demonstrate that the optical reflectivity of Nd_2CuO_4 contains a dynamical component with a timescale of 18 fs. I am struggling to establish the exact logic connecting this to "spin relaxation". My understanding is that the assertions are:

- Lightly doped cuprates are insulating due to "spin-polaron" effects up to a static carrier doping of $\sim 2\%$
- The reflectivity effect seems to change upon an effective transient doublon concentration of about 2% associating the 18 fs timescale with the insulating transition and the "spin-polarons"

My main problem with this line of reasoning is that I think it's incorrect to present the

transition as purely "spin-polaron". There can (and seemingly must) be a polaron associated with charge effects too. This is discussed a lot in the literature [e.g. PRL 95, 227002 (2005)]. While I agree that coherent long wavelength phonon effects might be slower than 18 fs, I don't see why local polaronic charge effects can't be very fast.

I would also note that reflectivity is fundamentally a probe of charge, so in the case of mixed character polarons, charge effects might be more likely to be seen.

Comment 3

Conclusion B:

I think this effect is nicely demonstrated. This not immediately obvious to me whether anything new has been learned regarding the cuprates. I would be interested in any further arguments or developments of the narrative in manuscript.

Small comments:

Comment 4

1. Page 3: this sentence implies the charge carriers are moving,

"In such cuprates, charge-spin coupling affects a charge carrier strongly, which moves on the antiferromagnetic spin background⁷⁻¹⁰."

A major point of the article is the spin-polaron effect, in which the carriers do not move.

Comment 5

2. Figure 1 is titled: "pump-probe reflection measurements", so I spent a few seconds looking for a pump-probe measurement, but what is meant here is pump-probe reflection setup

Comment 6

3. Figure 1 a) the spins are in the wrong direction.

Comment 7

4. In Fig 3f, maybe there is an unusual unit choice, but I usually expect to see the maximum of the dispersion at $(\pi,0)$.

Reply to the comments of the reviewer #1

We would like to thank the reviewer #1 for his/her careful reading of our paper, and valuable comments and suggestions. We would also like to appreciate his/her comment; “The research team achieved an outstanding time resolution of 10 fs, which is a major strength of the study. It is also reasonable to say that the behavior of Mott insulating cuprates is very important because they are often thought of as the canonical strongly correlated problem. The combination of these points put this study in good contention for Nat. Comms.”

We show the replies to his/her comments below, in which we divide his/her comments into 7 parts (comment 1-7) as shown above.

Comment 1 of the reviewer #1

Having said this, I think that Conclusion A is not sufficiently strongly established to qualify for Nat. Comms. Conclusion B is well-established, but I do not think there is a sufficiently strong case that something important about cuprates or ultra-fast optics has been learned here. I would be willing to re-assess this if the authors can expand on a stronger argument for conclusion A or a more compelling discussion of the importance of conclusion B.

Reply to comment 1 of the reviewer #1

The reviewer #1 raises the points A and B to be considered. We will answer the point A in the reply to comment 2 and answer the point B in the reply to comment 3, which are shown below.

Comment 2 of the reviewer #1

In order to start this conversation, I would emphasize that cuprates are well known to have strong charge-spin coupling and are very often discussed as having appreciable electron-phonon coupling. So one really needs to add detailed information about these processes rather than just show this as a general truth.

Conclusion A:

The authors adequately demonstrate that the optical reflectivity of Nd₂CuO₄ contains a dynamical component with a timescale of 18 fs. I am struggling to establish the exact logic connecting this to "spin relaxation". My understanding is that the assertions are:

- Lightly doped cuprates are insulating due to "spin-polaron" effects up to a static carrier

doping of ~2%

- The reflectivity effect seems to change upon an effective transient doublon concentration of about 2% associating the 18 fs timescale with the insulating transition and the "spin-polarons"

My main problem with this line of reasoning is that I think it's incorrect to present the transition as purely "spin-polaron". There can (and seemingly must) be a polaron associated with charge effects too. This is discussed a lot in the literature [e.g. PRL 95, 227002 (2005)]. While I agree that coherent long wavelength phonon effects might be slower than 18 fs, I don't see why local polaronic charge effects can't be very fast.

I would also note that reflectivity is fundamentally a probe of charge, so in the case of mixed character polarons, charge effects might be more likely to be seen.

Reply to comment 2 of the reviewer #1

The reviewer #1 suggested that the charge-phonon coupling is important as well as the charge-spin coupling in two-dimensional cuprates. Fundamentally, we agree with the reviewer #1. As the reviewer #1 commented, the effects of the charge-phonon coupling in undoped cuprates as well as doped ones have been indeed discussed in several previous studies.

First, let us discuss the possibility of the small polaron formation under the influence of charge-phonon coupling. Hereafter, we call it "*lattice polaron*". As seen in Fig. 2b, the sign of the additional reflectivity change (step B) with the rise time τ_r of 18 fs is negative and is the same as the sign of the initial decrease of the reflectivity (step A), that is, the main bleaching signal. Namely, in step B, the bleaching signal or the absolute value of the negative reflectivity change $|\Delta R/R|$ is enhanced up to $t \sim 20$ fs. This means that *photocarriers become more mobile with this time constant (the rise time of $\tau_r=18$ fs)*. If photocarriers are relaxed to lattice polarons due to the charge-phonon coupling in the same time domain, the itinerancy of carriers is necessarily suppressed by the surrounding displacements of ions. This should cause the reduction of the bleaching signal, resulting in the decrease of the absolute value of the negative reflectivity change $|\Delta R/R|$. The experimental result shows the opposite behavior, that is, the further decrease of the reflectivity in step B. Therefore, the experimental results cannot be explained by lattice-polaron (small-polaron) formations.

As discussed in the main text, under the presence of the antiferromagnetic spin background on a two-dimensional copper oxide plane, a doublon and a holon cannot

freely move. It is because a movement of the doublon or the holon necessarily disturbs the original antiferromagnetic spin arrangement. An introduced carrier (a doublon or a holon) itself will weaken the effective antiferromagnetic interactions between surrounding spins and slightly relax the original antiferromagnetic spin arrangement as shown in Fig. 4d. *This makes carriers a bit more delocalized and the bleaching signal due to the carriers is rather enhanced up to $t \sim 20$ fs.* Such a relaxed state is called a magnetic polaron in our paper, although that is considerably different from the lattice polaron or equivalently a small polaron under the influence of charge-phonon coupling. A magnetic polaron (a doublon or a holon after the spin relaxation) becomes a bit more mobile, but still cannot freely move on a copper oxide plane as discussed in a lot of previous papers.

The nature of photocarriers can also be discussed from the viewpoint of dynamics. We consider that the formation time of the lattice polaron should be related to the period of the phonon mode, which stabilizes the lattice polaron. The formation time is usually considered to be almost equal to a half of the period [e.g., A. Cavalleri *et al.*, PRB **70**, 161102(R) (2004)]. The highest frequency (energy) of optical phonon modes in Nd₂CuO₄ is ~ 540 cm⁻¹ (63 meV) [S. Tajima *et al.* PRB **43**, 10496 (1991)]. The period of the mode is ~ 60 fs. The half of the period ~ 30 fs is considerably longer than the time constant $\tau_r = 18$ fs of step B. The reviewer #2 suggested that the time scale of the lattice-polaron formation might depend on the magnitude of the charge-phonon coupling. We consider that the characteristic time constant cannot become shorter than the half period of the related phonon mode, since the spring constant of the local Cu-O bond is responsible for the lattice-polaron formation.

These two facts, i.e., the sign and the time constant of the reflectivity change in step B, exclude the contribution of the charge-phonon coupling to the carrier dynamics experimentally observed in our study.

Next, let us discuss why the photocarriers are not affected by the charge-phonon coupling so much in Nd₂CuO₄. The previous studies about the charge-phonon coupling in cuprates have been carried out in undoped and doped cuprates such as La₂CuO₄ and La_{2-x}Sr_xCuO₄, and the important role of the charge-phonon coupling on the carrier dynamics has been established in those compounds [e.g. O. Rösch *et al.*, PRL**95**, 227002 (2005)]. In contrast, few studies have been performed in Nd₂CuO₄. Several years ago, we have performed a comparative study on Nd₂CuO₄ and La₂CuO₄ using a pump-probe absorption spectroscopy with the time resolution of about 200 fs [H. Okamoto *et al.* PRB **83**, 125102 (2011)]. The results show that the photon energies of the mid-gap absorption

peaks due to the magnetic polarons (spin-relaxed photocarriers) observed in the mid-infrared region are lower in Nd_2CuO_4 than in La_2CuO_4 . This result was reasonably interpreted by the fact that the charge-phonon coupling is weaker in Nd_2CuO_4 than in La_2CuO_4 . Namely, in La_2CuO_4 , the magnetic polarons photogenerated are further localized by the effect of the charge-phonon coupling and the induced absorption of those carriers show blue shifts. Such an effect is relatively small in Nd_2CuO_4 . In addition, the recombination time of photocarriers in the case of the low carrier density is considerably shorter in Nd_2CuO_4 than in La_2CuO_4 . This means that the carrier motions are faster in Nd_2CuO_4 than in La_2CuO_4 . This result also indicated that the charge-phonon coupling is weaker in Nd_2CuO_4 than in La_2CuO_4 .

The difference in the magnitudes of the charge-phonon coupling in Nd_2CuO_4 and La_2CuO_4 is considered to originate from apical oxygen atoms existing in the copper oxide layers in La_2CuO_4 . They are not in Nd_2CuO_4 . It was indeed pointed out that the apical oxygen atoms play significant roles in the charge-phonon coupling, especially, in electrons. The details are reported in H. Okamoto *et al.* PRB **83**, 125102 (2011). The point in this reference is summarized as follows;

*“A theoretical study using a cluster model suggested that the charge-phonon coupling on a doped hole (a holon) due to the apical-oxygen mode is comparable to that due to the planar-oxygen mode [O. Rösch et al., PRL **95**, 227002 (2005)]. Therefore, a hole is more stabilized in La_2CuO_4 than in Nd_2CuO_4 . It is reasonable to consider that an electron (a doublon) is stabilized more strongly than a doped hole (a holon) by displacements of the apical oxygen atoms. It is because an electron carrier (a doublon) has Cu d character and can be directly affected by the displacements of apical oxygen atoms. Namely the site-diagonal-type charge-phonon coupling due to the apical oxygen atoms should play a dominant role in the stabilization of a doublon.”*

We think that the more definite information would be obtained by comparing the ultrafast responses with the time resolution of 10 fs in La_2CuO_4 and in Nd_2CuO_4 . However, in our experimental system, the tuning range of the output of the non-collinear optical parametric amplifier is limited, so that we cannot carry out the similar measurement in La_2CuO_4 . Note that the charge-transfer gap energy is larger in La_2CuO_4 than in Nd_2CuO_4 . Anyway, the comparative studies on La_2CuO_4 and Nd_2CuO_4 is an interesting problem, which should be done in future.

Taking the above discussions into account, we added the detailed description about the ultrafast dynamics in step B including the lattice polaron effect due to the charge-phonon coupling as follows.

[Line 9 of page 6 – Line 12 of page 7]

“First, we discuss the origin of the reflectivity change in step B, since it is an unusual dynamics. As mentioned above, it is natural to relate this step B to magnetic-polaron formation, while charge-phonon coupling should also be considered as well as charge-spin coupling in the formation of magnetic polaron. In fact, in the previous study of the photoemission spectroscopy on another typical undoped cuprate La_2CuO_4 , a polaronic behavior of a hole was identified²⁴. However, we can consider that the charge-phonon coupling does not play important roles on step B observed in Nd_2CuO_4 from the following reasons. The sign of the additional reflectivity change in step B is negative and the same as that of the initial decrease of the reflectivity (step A), that is, the main bleaching signal due to the photocarrier generation. Namely, in step B, the bleaching signal or the absolute value of the negative reflectivity change $|\Delta R/R|$ is enhanced up to $t \sim 20$ fs. This means that photocarriers become more itinerant with the rise time of $\tau_r = 18$ fs. If photocarriers are relaxed to small polarons due to the charge-phonon coupling (lattice polarons) in the same time domain, the itinerancy of carriers is suppressed by the surrounding displacements of ions. This should cause the reduction of the bleaching signal, resulting in the decrease of the absolute value of the negative reflectivity change $|\Delta R/R|$. The experimental result (step B) shows the opposite behavior, that is, the further decrease of the reflectivity. Therefore, the lattice-polaron formation cannot explain the experimental results. This interpretation is also supported by the time scale of optical phonons. The highest frequency (energy) of optical phonon modes in Nd_2CuO_4 is $\sim 540 \text{ cm}^{-1}$ (63 meV)²⁵. The period of that mode is ~ 60 fs, which is much longer than the time constant $\tau_r = 18$ fs in the response of step B. This fact also excludes the contribution of the charge-phonon coupling to step B. An introduction of a photocarrier itself necessarily relaxes the surrounding antiferromagnetic spin arrangement. That makes the carriers a bit more itinerant and therefore leads to the increase of the bleaching signal as observed in the experiments. Thus, we can conclude that the magnetic-polaron formation is responsible for step B, and $\tau_r = 18$ fs is the spin-relaxation time.”

Subsequently, we added the reason why the charge-phonon coupling in Nd_2CuO_4 is not so large.

[Line 13 – 19, page 7 in the new MS]

“In our results, no dynamical behavior originating from the charge-phonon coupling is observed. This is attributable to the fact that the charge-phonon coupling in Nd_2CuO_4 is relatively smaller than in La_2CuO_4 , in which the effect of the charge-phonon coupling on doped carriers was demonstrated. We consider that in Nd_2CuO_4 the charge-phonon

coupling is relatively smaller than in La_2CuO_4 mainly because apical oxygen atoms are absent in Nd_2CuO_4 . The role of apical oxygen atoms on the charge-phonon coupling was detailed in the previous studies^{16,17}”

To make clear the discussion, we also modified the related descriptions before the above discussions.

[Line 1 – 4, page 5 in the new MS]

“This bleaching signal reflects the dynamics of both the mid-gap-absorption and Drude components, since the spectral weight of the Mott-gap transition is transferred to the latter two components^{16,17}”

[Line 9 – 11, page 5 in the new MS]

“Therefore, it is natural to consider that step B is related not to the Drude component but to the magnetic-polaron formations.”

Comment 3 of the reviewer #1

Conclusion B:

I think this effect is nicely demonstrated. This not immediately obvious to me whether anything new has been learned regarding the cuprates. I would be interested in any further arguments or developments of the narrative in manuscript.

Reply to comment 3 of the reviewer #1

To clarify a role of the charge-spin coupling on a photocarrier in two-dimensional undoped cuprates is an important subject to be studied. It is because that is a fundamental information to solve complicated non-equilibrium dynamics in photoexcited Mott insulators. In fact, many theoretical studies about the charge-spin coupled dynamics after the photoexcitation in Mott insulators have recently been reported (e.g., refs. 7-11). We feel that a new scientific field, “non-equilibrium quantum mechanics in correlated electron systems” is now being developed. However, in the experimental side, no one has succeeded in real-time observation of spin dynamics in cuprates. In our study, we have really observed charge-spin-coupled dynamics for the first time not only in step B in the reflectivity change (Fig. 2) but also in the high-frequency oscillation on the spectrally resolved reflectivity changes (Fig. 3). The notable feature in the latter phenomenon is that the frequency of the oscillation depends on the probe energy, which can be explained by the interferences between charge excitations with two magnons. In addition, our results present an unambiguous evidence for the existence of magnon sideband just above the Mott gap transition.

We would also like to emphasize that the combination of the ultrafast pump-probe experiments and the analyses of the results to derive the charge-spin coupled dynamics would be a benchmark in the studies of photodoped cuprates, which we believe enables us to compare the experimental data and theoretical studies recently developed. This point has also been commented on by the reviewer #2,

To emphasize these points, we isolated a summary paragraph and modified it as follows.

[Line 16, page 11 – line 7, page 12]

“In summary, we have applied the pump-probe reflection spectroscopy with the time resolution of 10 fs on a typical Mott insulator of an undoped cuprate, Nd_2CuO_4 . We observed ultrafast responses originating from the charge-spin coupling in the photoexcited state. In the weak excitation case, we clearly identified the enhancement of the bleaching signal with the rise time of 18 fs just after the photocarrier generation, which was attributed to the time constant of the magnetic-polaron formation. In the spectrally resolved pump-probe experiments, we detected the high-frequency ($1400\text{--}2700\text{ cm}^{-1}$) coherent oscillation on the reflectivity changes. The oscillation frequency depends on the probe energy, which was explained by the interferences between charge excitations with two magnons. This is not only the unambiguous evidence that the charge-spin coupling generates a two-magnon sideband just above the Mott-gap transition peak but also the first real-time detection of charge-spin coupled dynamics in a Mott insulator of undoped cuprates. The result presented here would be a benchmark, which can be compared to modern theoretical calculations of non-equilibrium charge and spin dynamics in photoexcited Mott insulators.”

Comment 4 of the reviewer #1

1. Page 3: this sentence implies the charge carriers are moving,

"In such cuprates, charge-spin coupling affects a charge carrier strongly, which moves on the antiferromagnetic spin background⁷⁻¹⁰."

A major point of the article is the spin-polaron effect, in which the carriers do not move.

Reply to comment 4 of the reviewer #1

We would like to thank the reviewer #1 for his/her careful reading of our paper. The used phrase might be misleading, although photocarriers generated in the CuO planes are not completely localized as mentioned above. Taking the reviewer's comment into account, we modified the sentence in the introduction as follows.

[Line 2 – 4 of page 3 in the old MS]

“In such cuprates, charge-spin coupling affects a charge carrier strongly, which moves on the antiferromagnetic spin background⁷⁻¹⁰.”

→

[Line 2 – 4 of page 3 in the new MS]

"In such cuprates, charge-spin coupling strongly affects the dynamics of charge carriers on the antiferromagnetic spin background⁷⁻¹⁰."

Comment 5 of the reviewer #1

2. Figure 1 is titled: "pump-probe reflection measurements", so I spent a few seconds looking for a pump-probe measurement, but what is meant here is pump-probe reflection setup

Reply to comment 5 of the reviewer #1

We thank the reviewer #1 for his/her thoughtful comment. We modified the title of Fig. 1 as follows.

[Line 1 of page 17 in the old MS]

"pump-probe reflection measurements"

→

[Line 1 of page 20 in the new MS]

"pump-probe reflection setup"

Comment 6 of the reviewer #1

3. Figure 1 a) the spins are in the wrong direction.

Reply to comment 6 of the reviewer #1

We thank reviewer #1 for pointing out our mistake. We corrected the spin orientations in Figs. 1a and 4.

Comment 7 of the reviewer #1

4. In Fig 3f, maybe there is an unusual unit choice, but I usually expect to see the maximum of the dispersion at $(\pi,0)$.

Reply to comment 7 of the reviewer #1

Taking the comment of the reviewer #1 into account, we corrected Fig. 3f.

Comments of the reviewer #2

The manuscript by Miyamoto et al. presents an experimental investigation of the relaxation dynamics of photo-induced carriers in an undoped cuprate Mott insulator. A 7fs/2eV laser pulse is used to excite carriers across the gap and probe the subsequent bleaching of the charge-transfer (CT) resonance. The dynamics is analyzed in detail as function of the excitation density: For low photon number, an increase of the bleaching signal within 18fs after the excitation is observed, which is interpreted as a signature of the spin-polaron formation. When the fluence is increased, an additional fluence-dependent response is observed which reduces the bleaching; due to the fluence dependence, this can be interpreted as consequence of Auger recombination processes. Finally, an oscillatory component in the response at weak excitation density is interpreted as beating between CT-excitations with different magnon numbers.

A precise understanding of the carrier dynamics in Mott insulators is of great importance for the active research on materials out of equilibrium. The dynamics reflects the interaction of carriers and spins, which in turn is important for the superconducting pairing, so that a detailed knowledge of the ultra-fast carrier dynamics may help understanding light-induced superconductivity or superconductivity in the cuprates in general. In contrast to many other studies in the field, which focus on the doped material, the present work investigates the *undoped* cuprates. Only if the un-doped material is understood, there is a chance to shed light on the doped system. While the undoped material might not be as interesting regarding its material properties, I believe that the precise analysis as presented in the manuscript is a very important benchmark for the field: Because the properties of the undoped material are easier (but not easy!) to address theoretically, the study provides the basis for a detailed comparison to theory, which is crucially needed for further progress, but has rarely been attempted regarding the dynamics of strongly correlated systems in general. The observed timescale for spin-relaxation is in fact consistent with theoretical predictions (Ref. 7-10). Beyond that, the results of the manuscript can stimulate further studies: At present there is no systematic theoretical understanding including the excitation density dependence and hence the interaction between photo-doped carriers. Related open questions include the differentiation between p-holes and d-electrons and the p-d interaction. Furthermore, antiferromagnetic correlations in the 2d Hubbard model have recently been also studied in quantum simulation experiments with cold-atoms. The investigation of the short-time dynamics in the undoped cuprates seems to be a good candidate for a relatively clean setting in which a comparative study of experiments in these two fields may be envisaged.

I therefore believe that the topic is timely and of broad interest and thus suitable for publication in Nature Communications.

Besides this I have a number of questions/comments regarding the discussion of the results:

Comment 1

(1) It is possible to discuss the temperature-dependence of the results? Increasing temperature weakens the spin correlations and thus might have an effect on the relaxation process, which could give further support to the interpretation.

Comment 2

(2) Why can it be assumed in the analysis of the strong-fluence data that the signal from the spin-polaron formation is not fluence dependent? One could expect a saturation effect, i.e., for strong fluence the spin order is partially melted, so that the relaxation should slow down. Is it possible to give an estimate at which excitation density such a saturation effect might be expected?

Comment 3

(3) The beating between the zero and two-magnon-dressed states (Fig.3) is a nice interpretation of the data. How much does it rely on the sharpness of the CT excitation? The CT resonance seems to be broader than the beating frequency. On the other hand, as the beating occurs at the difference frequency between probe frequency and the CT resonance, should it not de-phase on the timescale set by the inverse width of the CT resonance?

Comment 4

(4) Assuming that the present interpretation of the beating signal is correct, can one discuss what could be learned from the measured signal? First, its amplitude could give information on the density of states of the modes which are involved. For the two-magnon states, there might be both a maximal energy (set by the maximum of the magnon dispersion), but possibly also a minimal energy (if the material has no long-range antiferromagnetic order, there are just paramagnons which do not exist at arbitrary low energy). On the other hand, the amplitude of the signal does not seem to depend on the frequency (Fig 3c)? Second, the de-phasing of the beating may give information on the lifetime of the coherence between the CT and the CT+2magnon state. In this case,

it may be interesting to discuss its dependence on the excitation density.

Comment 5

(5) For a further characterization of the beating signal it would be interesting to know how it scales with the amplitude of pump and probe. If it scales both linear in pump and probe in the weak excitation regime, it could be interpreted in terms of non-linear optics.

Comment 6

(6) This is a minor comment on the interpretation of process "B" in terms of spin-polaron formation: I agree that the interaction with lattice vibrations seems to be unlikely to be the origin of the process if the energy scale of optical phonons is smaller by a factor of three than the 220meV which correspond to 18fs. However, I think that in general such an argument should be used with care. The timescale on which energy is passed to some bosonic excitation is not simply related to the inverse of the energy of that bosonic excitation, but rather to the strength of its coupling to the electrons. The prime example is spin itself - the rate at which spin excitations are created in an antiferromagnet is set by the electron hopping, and the magnon energy only determines how much energy is transferred per spin excitation.

In summary, I find this is a nicely presented high-quality set of data which contribute to a very timely and important question in the field of ultra-fast dynamics in correlated systems. With a discussion that addresses the above points (as far as possible), I think the manuscript can be published in Nature Communications.

Reply to the comments of the reviewer #2

We would like to thank the reviewer #2 for his/her careful reading of our paper, and valuable comments and suggestions. We would also like to appreciate his/her comment; “I find this is a nicely presented high-quality set of data which contribute to a very timely and important question in the field of ultra-fast dynamics in correlated systems.”

We show the replies to his/her comments below, in which we divide his/her comments into 6 parts (comment 1-6) as shown above.

Comment 1 of the reviewer #2

(1) It is possible to discuss the temperature-dependence of the results? Increasing temperature weakens the spin correlations and thus might have an effect on the relaxation process, which could give further support to the interpretation.

Reply to comment 1 of the reviewer #2

We thank the reviewer #2 for his/her important comment. We agree with the point raised by the reviewer #2; the temperature dependence of the photo-responses should depend strongly on temperature. Considering the fact that the antiferromagnetic exchange interaction J is about 1800 K [P. Bourges *et al.* PRL**79**, 4906 (1997)], the sample temperature should be increases up to about 1000 K to detect the effect of the decrease in the spin-spin correlation on the carrier dynamics. To do this, a large heating furnace is usually used. In the pump-probe setup, such a large heating system cannot be introduced and therefore a small size of a heating system should be newly designed and constructed. At present, therefore, it is difficult to do such high temperature pump-probe experiments. We would like to consider the temperature dependence of photo-responses in the high temperature region as a future subject.

Comment 2 of the reviewer #2

(2) Why can it be assumed in the analysis of the strong-fluence data that the signal from the spin-polaron formation is not fluence dependent? One could expect a saturation effect, i.e., for strong fluence the spin order is partially melted, so that the relaxation should slow down. Is it possible to give an estimate at which excitation density such a saturation effect might be expected?

Reply to comment 2 of the reviewer #2

As the reviewer #2 pointed out, in the high fluence region, it is natural to consider that the spin relaxation time depends on the excitation fluence, since the antiferromagnetic

spin arrangement is largely modified depending on the excitation fluence. It is however difficult to clarify this point experimentally. In the case of the strong excitation, the ultrafast bleaching component, whose spectral weight is transferred to the Drude component, is enhanced as shown by the pink line in Fig. 2(b,c). The presence of this component makes it difficult to derive the contribution of the magnetic-polaron dynamics, and to clarify the excitation photon density dependence of the spin-relaxation time, that is, the time constant τ_r of the magnetic-polaron formation. Considering this point, in our study, therefore, we evaluated the time constant τ_r from the data in the low excitation fluence condition, in which only the response (step B) due to the magnetic-polaron formation appears in addition to the initial bleaching signal due to the photocarrier generation. As for the data in the higher excitation fluence region, we focused on deriving the excitation-fluence dependence of the Drude component. In the analysis of the higher excitation fluence data, we assumed that the time constant τ_r of the magnetic polaron formation is constant. We think that this assumption does not affect so much the evaluations of the excitation fluence dependence of the magnitude A_3 and the decay time τ_2 of the Drude component, since the Drude component shows apparently different behaviors from the spin relaxation processes of the magnetic polarons as shown in Figs. 2(b-d).

It is still an important issue to investigate the excitation fluence dependence of the spin dynamics. By performing the pump-probe measurement with high time resolution (10 fs) at the probe energies in the near-infrared to the mid-infrared region, in which the mid-gap absorption peaks appear, the spin relaxation processes might be directly detected. It is however very difficult to create such a short pulse in the infrared region, so that this is also a future problem.

Taking the comment 2 of the reviewer #2 into account, we added the following discussion.
[Line 17, page 8 – line 4, page 9]

“In the analyses for the medium- and strong-excitation cases, we used the same parameter values of τ_r and A_2/A_1 showing the time and the magnitude of the spin relaxation as mentioned above. This is a crude assumption, because with increase of x_{ph} the antiferromagnetic spin arrangement is further disturbed and τ_r might become long. In the analyses of the medium- and strong-excitation cases, however, it is important to evaluate the excitation photon density dependence of the magnitude A_3 and the decay time τ_2 of the Drude component. Even if the spin relaxation time τ_r becomes longer with increase of x_{ph} , the characteristic features of the x_{ph} dependence of A_3 and τ_2 shown in Figs. 2f and g, respectively, would not be affected so much, since the Drude

component shows apparently different behaviors from the spin relaxation processes of the magnetic polarons as shown in Figs. 2b-d.”

In addition, we modified the related sentences.

[Line 22 of page 6 – line 2 of page 7 in the old MS]

“The values of the parameters A_3 and τ_2 , which reflect the number of free carriers and their decay times, are plotted as a function of x_{ph} in Figs. 2f and g, respectively. The number of free carriers (A_3) shows a clear threshold behavior, which is characteristic of two-dimensional Mott insulators¹⁷.”

→

[Line 6 – 10 of page 8 in the new MS]

“The values of the parameters A_3 and τ_2 , which reflect the effective number of free carriers and their decay times, are plotted as a function of x_{ph} in Figs. 2f and g, respectively. The effective number of free carriers or the spectral weight of the Drude component (A_3) shows a clear threshold behavior, which is characteristic of two-dimensional Mott insulators¹⁷.”

[Line 14 – 16 of page 6 in the old MS]

“Upon considering the local nature of the spin relaxation processes, the analyses use the same values of τ_r and A_2/A_1 as those obtained for the weak excitation case.”

→

[Line 22 – 23 of page 7 in the new MS]

“In the analyses, we use the same values of τ_r and A_2/A_1 as those obtained for the weak excitation case.”

Comment 3 of the reviewer #2

(3) The beating between the zero and two-magnon-dressed states (Fig.3) is a nice interpretation of the data. How much does it rely on the sharpness of the CT excitation? The CT resonance seems to be broader than the beating frequency. On the other hand, as the beating occurs at the difference frequency between probe frequency and the CT resonance, should it not de-phase on the timescale set by the inverse width of the CT resonance?

Reply to comment 3 of the reviewer #2

As the reviewer #2 pointed out, the dephasing time of the quantum beat between two kinds of excitation paths is likely to depend on the dephasing time of the CT excited state.

We thank the reviewer #2 for his/her significant and thoughtful comment.

In our simple model presented in the discussion section in the main text, we consider a discrete excited state, which might correspond to an excitonic state. In undoped cuprates, the importance of the excitonic effect was pointed out (e.g., ref. 15), while the magnitude of the excitonic effect or the exciton binding energy has not been evaluated quantitatively. The optical conductivity shows the rather flat spectral shape in the higher energy side of the peak at 1.6 eV. It is natural to relate this to the electron-hole continuum. Namely, the lowest CT exciton state seems to be located close to the bottom edge of the continuum. Since other undoped cuprates such as La_2CuO_4 and $\text{Sr}_2\text{CuO}_2\text{Cl}_2$ show similar optical conductivity spectra, the weak excitonic effect seems to be a common feature in undoped cuprates. In this situation, it is difficult to evaluate precisely the spectral width of the lowest CT exciton. By fitting the lower half of the Lorentzian spectrum to the experimental spectrum below 1.7 eV, we roughly estimated the half width at half maximum to be 0.18 eV. The full width at half maximum Γ is 0.36 eV. Using a simple formula, $\tau_{\text{phase}} = \frac{1}{\pi\Gamma}$ (τ_{phase} : the dephasing time), we obtain $\tau_{\text{phase}} \sim 3.5$ fs.

In our experiments, the oscillation is definitely observed, but it is difficult to derive the oscillatory component near the time origin $t = 0$, since the initial bleaching signal sharply enhanced around $t = 0$. Therefore, we showed in Fig. 3 the time evolution of the oscillation only after 20 fs. From the time characteristics shown in Fig. 3b, however, the decay time τ_{decay} of the oscillation seems to be much longer than $\tau_{\text{phase}} \sim 3.5$ fs.

Considering those discussions, we speculate that the decay time of two magnons itself would be responsible for the decay time of the oscillatory components, although we do not have a clear explanation. To solve this issue is important to fully understand the dynamics of photoexcited states under the influence of charge-spin coupling in 2D Mott insulators. We expect that our data will be able to stimulate theoretical studies on this issue.

Comment 4 of the reviewer #2

(4) Assuming that the present interpretation of the beating signal is correct, can one discuss what could be learned from the measured signal? First, its amplitude could give information on the density of states of the modes which are involved. For the two-magnon states, there might be both a maximal energy (set by the maximum of the magnon dispersion), but possibly also a minimal energy (if the material has no long-range antiferromagnetic order, there are just paramagnons which do not exist at arbitrary low energy). On the other hand, the amplitude of the signal does not seem to depend on the

frequency (Fig 3c)? Second, the de-phasing of the beating may give information on the lifetime of the coherence between the CT and the CT+2magnon state. In this case, it may be interesting to discuss its dependence on the excitation density.

Reply to comment 4 of the reviewer #2

We thank the reviewer #2 for his/her constructive comments. We think that all of the points raised by the reviewer #2 are interesting. It is important for us to derive hidden information about dynamical aspects of photoexcited states from our data. We will try to answer them below, although most of questions given by the reviewer #2 have not been solved at the present stage.

It is interesting to evaluate the probe energy dependence of the amplitude of the oscillatory components as well as that of the decay time. The former might be related to the magnon density of state. Unfortunately, in the present setup, such an evaluation is technically difficult. We performed pump-probe measurement with the time resolution of 10 fs. However, the oscillatory frequencies associated with magnons are very high. Note that the period of the oscillation with the frequency of 2700 cm^{-1} observed at the probe energy of 2.10 eV is 12.4 fs, which is comparable to our time resolution. In this situation, the amplitudes of the oscillation experimentally measured are modified depending on the relative magnitude of the oscillation period and the time resolution.

In addition, a probe pulse after passing through a band-pass filter still has a finite spectral width ($\sim 10\text{ nm}$) and its spectral shape depends on the original spectrum of the pulse shown in Fig. 1c and the transmission characteristic of the filter. In this case, oscillations produced within the spectrum of a probe pulse are superimposed, which will modify not only the amplitude but also the decay time of an oscillatory component experimentally observed. These technical issues make it difficult to perform detailed quantitative analyses of the oscillations and evaluate their amplitudes and the decay times. On the other hand, the central frequency of each oscillatory component can be accurately determined directly from the experimental data. Therefore, we focused on it in the present study.

As suggested by the reviewer #2, it is also interesting to investigate the excitation photon-density dependence of the decay time of the oscillatory components. In the present study, we measured the oscillations in the low excitation photon-density region, since a coherent oscillation is usually suppressed with increase of the excitation photon density. We have not performed a systematic study about the excitation photon-density

dependence of the oscillatory signals yet. This is also a significant problem, which should be clarified in future.

Taking comment 4 of the reviewer #2 into account, we added a following description saying that it is difficult to discuss the probe energy dependence of the oscillation amplitudes.

[Line 21, page 9 – line 9, page 10]

“The amplitudes of the oscillations are also expected to include important information about the charge-spin coupling. However, in our experimental condition, it is difficult to compare precisely the relative oscillation amplitudes detected at different probe energies, since the oscillation frequencies are very high. The period of the oscillation with the frequency of 2700 cm^{-1} observed at the probe energy of 2.10 eV is 12.4 fs, which is comparable to the time resolution of 10 fs in our PP system. In such a situation, the amplitudes of the oscillation experimentally measured are suppressed with increase of the oscillation frequency. On the other hand, the central frequency of each oscillatory component can be accurately determined directly from the experimental data. So, we focused on it in the present study. The origin of the observed oscillations is discussed in detail in the next section.”

Comment 5 of the reviewer #2

(5) For a further characterization of the beating signal it would be interesting to know how it scales with the amplitude of pump and probe. If it scales both linear in pump and probe in the weak excitation regime, it could be interpreted in terms of non-linear optics.

Reply to comment 5 of the reviewer #2

In general, the fluence of a probe pulse should be weakened in a pump-probe experiment, so that no optical nonlinearity originating from the probe pulse itself appear. In addition, we used the low excitation photon density $x_{\text{ph}} = 0.008 \text{ ph/Cu}$ to detect the oscillatory component and the fluence of the probe pulse was at least one order smaller than that of the pump pulse.

We investigated the x_{ph} dependence of the reflectivity changes $\Delta R(t)/R$ (not monochromatized) as shown in Fig. 2, but did not investigate systematically that of the oscillatory component, $\Delta R_{\text{OSC}}(t)/R$, obtained after the probe pulse is monochromatized. For $x_{\text{ph}} < 0.008 \text{ ph/Cu}$, it is difficult to evaluate the amplitude of the oscillatory components due to the low S/N ratio. In the weak excitation case $x_{\text{ph}} \leq 0.008 \text{ ph/Cu}$, we assumed that the oscillatory component $\Delta R_{\text{OSC}}(t)/R$ is proportional to x_{ph} and

does not depend on the probe photon density. This is just a framework of the third-order optical nonlinearity. In fact, the probe energy dependence of the oscillation frequency can be explained by the model calculation, in which a framework of the third-order optical nonlinearity is adopted.

Comment 6

(6) This is a minor comment on the interpretation of process "B" in terms of spin-polaron formation: I agree that the interaction with lattice vibrations seems to be unlikely to be the origin of the process if the energy scale of optical phonons is smaller by a factor of three than the 220 meV which correspond to 18 fs. However, I think that in general such an argument should be used with care. The timescale on which energy is passed to some bosonic excitation is not simply related to the inverse of the energy of that bosonic excitation, but rather to the strength of its coupling to the electrons. The prime example is spin itself - the rate at which spin excitations are created in an antiferromagnet is set by the electron hopping, and the magnon energy only determines how much energy is transferred per spin excitation.

Reply to comment 6 of the reviewer #2

We would like to thank the reviewer #2 for his/her valuable comment about the interpretation of process "B".

The rise time τ_r of the additional decrease of the reflectivity cannot be attributed to any lattice dynamics, while it is reasonable attributable to the spin relaxation. The discussions about this point is shown in the reply to comment 1 of the reviewer #1. We would like to ask the reviewer #2 to see that.

Reviewers' comments:

Reviewer #1 (Remarks to the Author):

I would like to thank the authors for their additional effort in explaining their work and reiterate that I think the work has many positive points.

Comments 3-7 inclusive have been addressed adequately.

I feel like I have to return to Comments 1&2 which relate to the assignment of the 18 fs timescale to the spin-relation.

(i) I think it is reasonable to say that the timescale is a bit faster than one expects for a process involving a coherent long-wavelength phonon, but I did not find a compelling argument that 18 fs cannot possibly involve a local type of polaronic effect.

(ii) I also wonder whether charge-charge scattering processes cannot play a role? I would be happier if the timescale was independent of fluence. We see that the timescale is weakly fluence dependent, which is promising but not quite the definitive result one might hope for in an ideal world

In summary, I think the spin assignment is sensible, but I see a mismatch between the very strong and confident statements in the abstract and title and the results presented. Reading the abstract, one expects to find a very direct and unambiguous spin measurements such as Raman /RIXS etc. For these reasons, I feel like I cannot personally support publication in this form. I am a little reluctant to say this, as I recognize that the work is in many respects very nice. I would be open to an abstract that better conveys the results obtained or more solid data/arguments. As is, I feel that it is a marginal case whether the work belongs in Nature Communications or not.

Comment 8:

Please can the authors recheck the maximum phonon energy in Nd₂CuO₄. Neutron work seems to report significantly higher values Physica B 174 (1991) 323-329.

Reviewer #2 (Remarks to the Author):

After reading the response of the authors to the referee comments, I believe that the questions have been clarified. I agree that some of my suggestions/questions go beyond the scope of the present manuscript (i.e., the question whether one can use the two-magnon beating signal to further characterise the (para)magnons). I would also think that the electron-phonon coupling, addressed by referee #1, is an important issue, but that the current manuscript presents convincing evidence that the electron spin coupling plays a dominant role here.

In summary, I can repeat my judgement of the previous report, that this manuscript present an important contribution to the very active field of ultra-fast dynamics in correlated systems, and I recommend the manuscript for publication in Nature Communications.

Response to Reviewers

Summary of changes

The changed parts were colored in yellow in Marked MS2.pdf.

Comments of the reviewer #1

I would like to thank the authors for their additional effort in explaining their work and reiterate that I think the work has many positive points.

Comments 3-7 inclusive have been addressed adequately.

I feel like I have to return to Comments 1&2 which relate to the assignment of the 18 fs timescale to the spin-relation.

Comment (A)

(i) I think it is reasonable to say that the timescale is a bit faster than one expects for a process involving a coherent long-wavelength phonon, but I did not find a compelling argument that 18 fs cannot possibly involve a local type of polaronic effect.

Comment (B)

(ii) I also wonder whether charge-charge scattering processes cannot play a role? I would be happier if the timescale was independent of fluence. We see that the timescale is weakly fluence dependent, which is promising but not quite the definitive result one might hope for in an ideal world

Comment (C)

In summary, I think the spin assignment is sensible, but I see a mismatch between the very strong and confident statements in the abstract and title and the results presented. Reading the abstract, one expects to find a very direct and unambiguous spin measurements such as Raman /RIXS etc. For these reasons, I feel like I cannot personally support publication in this form. I am a little reluctant to say this, as I recognize that the work is in many respects very nice. I would be open to an abstract that better conveys the results obtained or more solid data/arguments. As is, I feel that it is a marginal case whether the work belongs in Nature Communications or not.

Comment (D)

Comment 8:

Please can the authors recheck the maximum phonon energy in Nd₂CuO₄. Neutron work seems to report significantly higher values Physica B 174 (1991) 323-329.

Reply to the comments of the reviewer #1

We would like to thank the reviewer #1 for his/her valuable comments and suggestions. We show the replies to his/her comments below, in which we divide his/her comments into 4 parts (comments A-D) as shown above.

Comment (A) of the reviewer #1

I think it is reasonable to say that the timescale is a bit faster than one expects for a process involving a coherent long-wavelength phonon, but I did not find a compelling argument that 18 fs cannot possibly involve a local type of polaronic effect.

Reply to comment (A) of the reviewer #1

The reviewer #1 pointed out that the time scale of the local type polaronic effect is not necessarily the same as the half of the period of the long-wavelength phonon. However, we cannot ascribe the observed time constant (18 fs) to the formation of lattice polarons from the following discussions, the main part of which was written in “*Reply to comment 2 of the reviewer #1*” in the previous reply.

First, let us discuss the possibility of the small polaron formation under the influence of charge-phonon coupling. Hereafter, we call it “*lattice polaron*”. As seen in Fig. 2b, the sign of the additional reflectivity change (step B) with the rise time τ_r of 18 fs is negative and is the same as the sign of the initial decrease of the reflectivity (step A), that is, the main bleaching signal. Namely, in step B, the bleaching signal or the absolute value of the negative reflectivity change $|\Delta R/R|$ is enhanced up to $t \sim 20$ fs. This means that *photocarriers become more mobile with this time constant (the rise time of $\tau_r=18$ fs)*. If photocarriers are relaxed to lattice polarons due to the charge-phonon coupling in the same time domain, the itinerancy of carriers should be suppressed by the surrounding displacements of ions. This causes the reduction of the bleaching signal, resulting in the decrease of the absolute value of the negative reflectivity change $|\Delta R/R|$. The experimental result shows the opposite behavior, that is, the further decrease of the reflectivity in step B. Therefore, the experimental results cannot be explained by lattice-polaron (small-polaron) formations.

As discussed in the main text, under the presence of the antiferromagnetic spin background on a two-dimensional copper oxide plane, a doublon and a holon cannot freely move. It is because a movement of the doublon or the holon necessarily disturbs the original antiferromagnetic spin arrangement. An introduced carrier (a doublon or a

holon) itself will weaken the effective antiferromagnetic interactions between surrounding spins and slightly relax the original antiferromagnetic spin arrangement as shown in Fig. 4d. *This makes carriers a bit more delocalized and the bleaching signal due to the carriers is rather enhanced up to $t \sim 20$ fs.* Such a relaxed state is called a magnetic polaron in our paper, although that is considerably different from the lattice polaron under the influence of charge-phonon coupling. A magnetic polaron (a doublon or a holon after the spin relaxation) becomes a bit more mobile, but still cannot freely move on a copper oxide plane as discussed in a lot of previous papers.

These discussions are summarized at line 9 of page 6 – line 12 of page 7 in the main text as follows.

“First, we discuss the origin of the reflectivity change in step B, since it is an unusual dynamics. As mentioned above, it is natural to relate this step B to magnetic-polaron formation, while charge-phonon coupling should also be considered as well as charge-spin coupling in the formation of magnetic polaron. In fact, in the previous study of the photoemission spectroscopy on another typical undoped cuprate La_2CuO_4 , a polaronic behavior of a hole was identified²⁴. However, we can consider that the charge-phonon coupling does not play important roles on step B observed in Nd_2CuO_4 from the following reasons. The sign of the additional reflectivity change in step B is negative and the same as that of the initial decrease of the reflectivity (step A), that is, the main bleaching signal due to the photocarrier generation. Namely, in step B, the bleaching signal or the absolute value of the negative reflectivity change $|\Delta R/R|$ is enhanced up to $t \sim 20$ fs. This means that photocarriers become more itinerant with the rise time of $\tau_r = 18$ fs. If photocarriers are relaxed to small polarons due to the charge-phonon coupling (lattice polarons) in the same time domain, the itinerancy of carriers is suppressed by the surrounding displacements of ions. This should cause the decrease of the bleaching signal, resulting in the decrease of the absolute value of the negative reflectivity change $|\Delta R/R|$. The experimental result (step B) shows the opposite behavior, that is, the further decrease of the reflectivity. Therefore, the lattice-polaron formation cannot explain the experimental results. This interpretation is also supported by the time scale of optical phonons. The highest frequency (energy) of the phonon dispersion curves in Nd_2CuO_4 is ~ 18 THz (74 meV)²⁵. The period of that mode is ~ 56 fs, which is much longer than the time constant $\tau_r = 18$ fs in the response of step B. This fact also excludes the contribution of the charge-phonon coupling to step B. An introduction of a photocarrier itself necessarily relaxes the surrounding antiferromagnetic spin arrangement. That makes the carriers a bit more itinerant and therefore leads to the increase of the bleaching signal as observed in

the experiments. Thus, we can conclude that the magnetic-polaron formation is responsible for step B, and $\tau_r = 18$ fs is the spin-relaxation time.”

As for the values colored in yellow in the above discussions, please see the reply to comment (D) of the reviewer #1.

Comment (B) of the reviewer #1

I also wonder whether charge-charge scattering processes cannot play a role? I would be happier if the timescale was independent of fluence. We see that the timescale is weakly fluence dependent, which is promising but not quite the definitive result one might hope for in an ideal world

Reply to comment (B) of the reviewer #1

We consider that it depends on the excitation fluence (or equivalently the photocarrier density) whether charge-charge scattering processes play some roles on the dynamics of photoexcited states.

In the case that the excitation photon density is low ($x_{\text{ph}} < 0.005$ ph/Cu), the time evolution of the reflectivity change is analyzed by the following formula.

$$\frac{\Delta R(t)}{R} = - \left\{ A_1 + A_2 \left[1 - \exp\left(-\frac{t}{\tau_r}\right) \right] \right\} \exp\left(-\frac{t}{\tau_1}\right) \quad (1)$$

The first term shows the bleaching signal due to the photocarrier generation and the second term is assigned to the spin-polaron formation. As mentioned in the reply to comment (A) of the reviewer #1, this term cannot be related to lattice polarons. In the weak excitation condition ($x_{\text{ph}} < 0.005$ ph/Cu), the time constant ($\tau_r = 18$ fs) is unchanged. This fact indicates that this relaxation process characterized by $\tau_r = 18$ fs can be attributed to an isolated photoexcited state (a pair of doublon and holon) and cannot be ascribed to carrier-density-dependent phenomena such as charge-charge scatterings. In the weak excitation case, photocarriers cannot move freely. In addition, photocarriers do not have large excess energies, since the excitation photon energy is just above the Mott-gap energy. Therefore, photocarriers cannot excite other carriers beyond the Mott gap. From these considerations, in the weak excitation case, we can neglect the effect of the charge-charge scattering.

In the analyses of the reflectivity changes $\Delta R(t)/R$ in the medium and strong excitation cases (Figs. 2c and d, respectively), we add the term $-A_3 \exp(-t/\tau_2)$ to equation (1) to express the Drude component with decay time τ_2 (step B'). In the analyses in these excitation conditions, we also assume that the spin-relaxation time $\tau_r =$

18 fs is constant. In the previous report, the reviewer #2 questioned this point, and we answered his/her question in *Reply to comment 2 of the reviewer #2* in the previous reply as follows.

“As the reviewer #2 pointed out, in the high fluence region, it is natural to consider that the spin relaxation time depends on the excitation fluence, since the antiferromagnetic spin arrangement is largely modified depending on the excitation fluence. It is however difficult to clarify this point experimentally. In the case of the strong excitation, the ultrafast bleaching component, whose spectral weight is transferred to the Drude component, is enhanced as shown by the pink line in Fig. 2(b,c). The presence of this component makes it difficult to derive the contribution of the magnetic-polaron dynamics, and to clarify the excitation photon density dependence of the spin-relaxation time, that is, the time constant τ_r of the magnetic-polaron formation. Considering this point, in our study, therefore, we evaluated the time constant τ_r from the data in the low excitation fluence condition, in which only the response (step B) due to the magnetic-polaron formation appears in addition to the initial bleaching signal due to the photocarrier generation. As for the data in the higher excitation fluence region, we focused on deriving the excitation-fluence dependence of the Drude component. In the analysis of the higher excitation fluence data, we assumed that the time constant τ_r of the magnetic polaron formation is constant. We think that this assumption does not affect so much the evaluations of the excitation fluence dependence of the magnitude A_3 and the decay time τ_2 of the Drude component, since the Drude component shows apparently different behaviors from the spin relaxation processes of the magnetic polarons as shown in Figs. 2(b-d).

It is still an important issue to investigate the excitation fluence dependence of the spin dynamics. By performing the pump-probe measurement with high time resolution (10 fs) at the probe energies in the near-infrared to the mid-infrared region, in which the mid-gap absorption peaks appear, the spin relaxation processes might be directly detected. It is however very difficult to create such a short pulse in the infrared region, so that this is also a future problem.”

These discussions are summarized at line 17, page 8 – line 4, page 9 in the main text.

“In the analyses for the medium- and strong-excitation cases, we used the same parameter values of τ_r and A_2/A_1 showing the time and the magnitude of the spin relaxation as mentioned above. This is a crude assumption, because with increase of x_{ph} the antiferromagnetic spin arrangement is further disturbed and τ_r might become long. In the analyses of the medium- and strong-excitation cases, however, it is important to

evaluate the excitation photon density dependence of the magnitude A_3 and the decay time τ_2 of the Drude component. Even if the spin relaxation time τ_r becomes longer with increase of x_{ph} , the characteristic features of the x_{ph} dependence of A_3 and τ_2 shown in Figs. 2f and g, respectively, would not be affected so much, since the Drude component shows apparently different behaviors from the spin relaxation processes of the magnetic polarons as shown in Figs. 2b-d.”

Comment (C) of the reviewer #1

In summary, I think the spin assignment is sensible, but I see a mismatch between the very strong and confident statements in the abstract and title and the results presented. Reading the abstract, one expects to find a very direct and unambiguous spin measurements such as Raman /RIXS etc. For these reasons, I feel like I cannot personally support publication in this form. I am a little reluctant to say this, as I recognize that the work is in many respects very nice. I would be open to an abstract that better conveys the results obtained or more solid data/arguments. As is, I feel that it is a marginal case whether the work belongs in Nature Communications or not.

Reply to comment (C) of the reviewer #1

We would like to thank the reviewer #1 for his/her thoughtful comment. Taking this comment, we modified the abstract. We believe that the modified abstract properly summarizes the content of our paper.

[Abstract in page 1] (The sentences colored in yellow were modified.)

“A charge excitation in a two-dimensional Mott insulator is strongly coupled with the surrounding spins, which is observed as magnetic-polaron formations of doped carriers and a magnon sideband in the Mott-gap transition spectrum. However, the dynamics related to the spin sector are difficult to measure. Here, we show that pump-probe reflection spectroscopy with 7-fs laser pulses can detect the optically induced spin dynamics in Nd_2CuO_4 , a typical cuprate Mott insulator. The bleaching signal at the Mott-gap transition is enhanced at ~ 18 fs. This time constant is attributable to the spin-relaxation time in magnetic-polaron formations, which is characterized by the exchange interaction. More importantly, ultrafast coherent oscillations appear in the time evolutions of the reflectivity changes, and their frequencies ($1400\text{--}2700\text{ cm}^{-1}$) are equal to the probe energy measured from the Mott-gap transition peak. These oscillations can be interpreted as the interferences between charge excitations with two magnons originating from charge-spin coupling.”

Comment (D) of the reviewer #1

Please can the authors recheck the maximum phonon energy in Nd₂CuO₄. Neutron work seems to report significantly higher values Physica B 174 (1991) 323-329.

Reply to comment (D) of the reviewer #1

We would like to thank the reviewer #1 for his/her variable comment. The maximum energy of the phonon dispersion curves of Nd₂CuO₄ measured by the inelastic neutron scattering experiment in the reference he/she introduced [Physica B **174**, 323-329 (1991).] is ~18 THz (~74 meV). The value of ~63 meV mentioned in our previous manuscript is of the transverse optical mode, which is directly measured in the optical reflectivity spectrum. In the new manuscript, we used the value of ~18 THz (~74 meV) as the highest frequency of the phonon modes in Nd₂CuO₄.

[Line 4-7 of page 7 in the old MS]

“The highest frequency (energy) of optical phonon in Nd₂CuO₄ is ~540 cm⁻¹ (63 meV)²⁵. The period of that mode is ~60 fs, which is much longer than the time constant $\tau_r = 18$ fs in the response of step B.”

→

[Line 4-7 of page 7 in the new MS]

“The highest frequency (energy) of the phonon dispersion curves in Nd₂CuO₄ is ~18 THz (74 meV)²⁵. The period of that mode is ~56 fs, which is much longer than the time constant $\tau_r = 18$ fs in the response of step B.”

We also added the related reference in the new manuscript.

25. Pintschovius, L. *et al.* Lattice dynamical studies of HTSC materials. *Physica B* **174**, 323-329 (1991).

Instead, we have deleted the reference 25 in the old manuscript.

As detailed in the reply to comment (A) of the reviewer #1, *the bleaching signal is enhanced with time up to ~20 fs*, which cannot be ascribed to the charge-phonon coupling irrespective of the τ_r value.